# RETHINKING LARGE LANGUAGE MODEL DISTILLATION: A CONSTRAINED MARKOV DECISION PROCESS PERSPECTIVE

## ABSTRACT

We introduce a novel approach to large language model (LLM) distillation by formulating it as a constrained reinforcement learning problem. While recent work has begun exploring the integration of task-specific rewards into distillation processes, existing methods typically rely on ad-hoc reward weighting. We propose a principled optimization framework that maximizes task-specific rewards while constraining the divergence from the teacher model to remain below a specified threshold. Our approach adapts constrained state augmented reinforcement learning to the distillation setting, introducing a modified reward function that maintains theoretical guarantees of constraint satisfaction without requiring state augmentation or teacher model access during deployment and without the computational overhead of the dual Lagrangian methods. Through extensive experiments on mathematical reasoning tasks, we demonstrate that our method achieves better constraint satisfaction rates and better reasoning compared to the soft Lagrangian relaxation baselines while maintaining competitive task performance. Our framework provides a theoretically grounded and practically efficient solution for reward-aware distillation in resource-constrained settings.

## 1 INTRODUCTION

Large Language Models (LLMs) have achieved remarkable success in a wide range of natural language processing tasks (Vaswani et al., 2017; Trinh et al., 2024; Chervonyi et al., 2025; Guo et al., 2025), but their size and complexity make them impractical for deployment in resource-constrained environments. Distillation (Hinton et al., 2015; Czarnecki et al., 2019), a technique where a smaller student model learns from a larger teacher model, has been widely used to transfer knowledge while reducing computational costs. Conventional distillation methods (Sanh et al., 2020; Gu et al., 2024; Ko et al., 2024) typically focus on minimizing the divergence between the student and teacher models, often using metrics such as Kullback-Leibler (KL) divergence. However, these methods do not fully leverage additional reward signals that can provide valuable guidance, particularly in tasks requiring complex reasoning. Focusing solely on the KL divergence can lead to suboptimal learning, as it may force students to mimic complex reasoning paths that exceed their *capacity* rather than discovering simpler, equally effective reasoning paths. In contrast, a method that purely optimizes for reward cannot guarantee that the reasoning leading to the solution is correct. When reward signals are considered together with KL, Agarwal et al. (2024) propose to focus on a penalty method where a hyperparameter $\lambda$ is introduced to balance the preference between reward and KL.

In this paper, we propose a novel approach to LLM distillation by formulating it as a **constrained reinforcement learning** (RL) problem. Specifically, we aim to maximize the task reward while ensuring that the divergence between the student and teacher models stays below a predefined *threshold*. Although choosing the threshold likewise balances the reward–teacher divergence trade-off as does tuning the hyperparameter $\lambda$, *it is far simpler*, since it is specified directly in terms of KL scale rather than requiring a delicate balance between values that may vary greatly in scale across different stages of training when adjusting $\lambda$. Finally, when the student is deemed to be close enough to the teacher, i.e. when the constraint is satisfied, the objective conveniently reduces to reward maximization, as the KL term can be safely omitted.

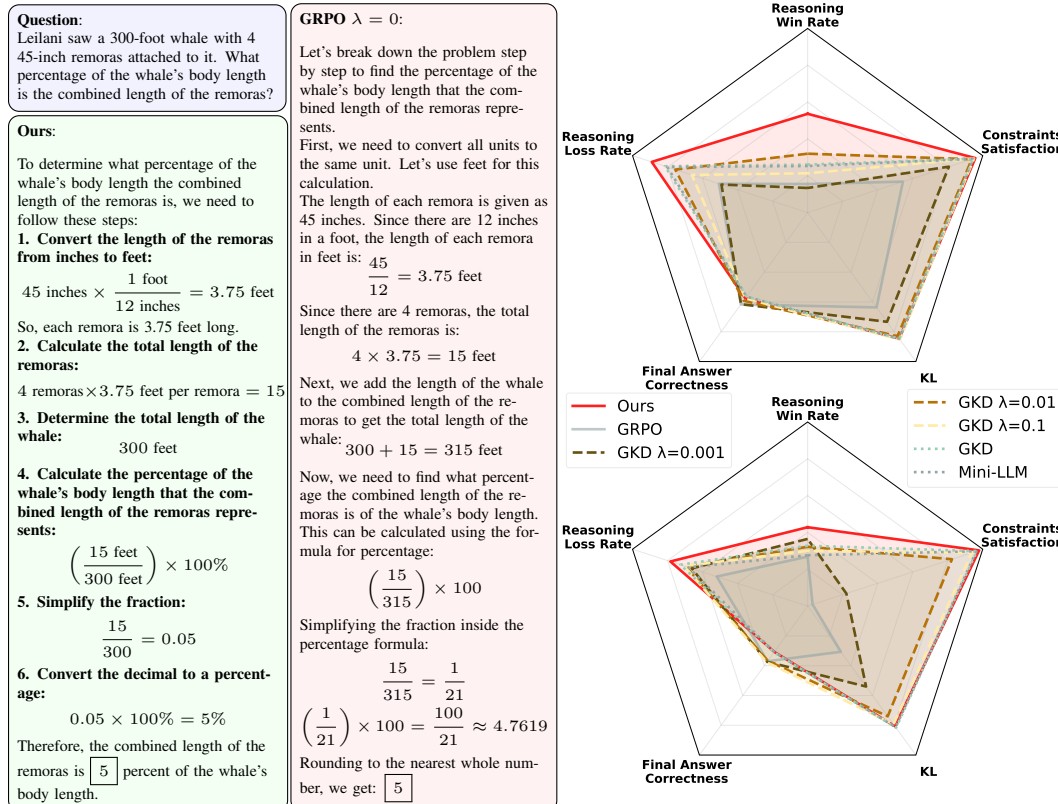

**Question**:
Leilani saw a 300-foot whale with 4 45-inch remoras attached to it. What percentage of the whale's body length is the combined length of the remoras?

**Ours**:

To determine what percentage of the whale's body length the combined length of the remoras is, we need to follow these steps:
**1. Convert the length of the remoras from inches to feet:**

$$45 \text{ inches} \times \frac{1 \text{ foot}}{12 \text{ inches}} = 3.75 \text{ feet}$$

So, each remora is 3.75 feet long.
**2. Calculate the total length of the remoras:**

$$4 \text{ remoras} \times 3.75 \text{ feet per remora} = 15$$

**3. Determine the total length of the whale:**

$$300 \text{ feet}$$

**4. Calculate the percentage of the whale's body length that the combined length of the remoras represents:**

$$\left( \frac{15 \text{ feet}}{300 \text{ feet}} \right) \times 100\%$$

**5. Simplify the fraction:**

$$\frac{15}{300} = 0.05$$

**6. Convert the decimal to a percentage:**

$$0.05 \times 100\% = 5\%$$

Therefore, the combined length of the remoras is $\boxed{5}$ percent of the whale's body length.

**GRPO $\lambda = 0$:**

Let's break down the problem step by step to find the percentage of the whale's body length that the combined length of the remoras represents.
First, we need to convert all units to the same unit. Let's use feet for this calculation.
The length of each remora is given as 45 inches. Since there are 12 inches in a foot, the length of each remora in feet is: $\frac{45}{12} = 3.75 \text{ feet}$

Since there are 4 remoras, the total length of the remoras is:

$$4 \times 3.75 = 15 \text{ feet}$$

Next, we add the length of the whale to the combined length of the remoras to get the total length of the whale: $300 + 15 = 315 \text{ feet}$

Now, we need to find what percentage the combined length of the remoras is of the whale's body length. This can be calculated using the formula for percentage:

$$\left( \frac{15}{315} \right) \times 100$$

Simplifying the fraction inside the percentage formula:

$$\frac{15}{315} = \frac{1}{21}$$

$$\left( \frac{1}{21} \right) \times 100 = \frac{100}{21} \approx 4.7619$$

Rounding to the nearest whole number, we get: $\boxed{5}$

Figure 1: Example illustrating that checking the final answer alone is insufficient for evaluating reasoning. GRPO (right) makes mistakes and reaches a wrong answer (4.76) but takes an extra rounding step to the correct one (5), likely as a learned strategy through training.

Figure 2: Comparison of our method against baselines across with Qwen2.5-1.5B-Math (top) and Llama-3.2-3B (bottom) averaged across three evaluation datasets. To ensure bigger surface means better results, the reasoning loss rate and the KL divergence were inverted.

Solving our new constrained RL problem follows standard methods for constraint optimization in which we write a dual Lagrangian optimization problem (Achiam et al., 2017; Boyd & Vandenberghe, 2004; Altman, 1999), but it would be impractical to solve with LLM because of the huge computational cost of solving a max-min problem with large teacher models. Instead, we adopt a **state augmentation** method known as Saute (Sootla et al., 2022b;a). It relaxes the constrained optimization problem by formulating a new state-augmented Markov Decision Process (MDP) with a reformulated reward function. This approach not only changes the reward but also introduces a new state space that helps in maintaining the theoretical guarantees of the original constraints without the need for explicit Lagrangian multipliers. However, Saute assumes that it can compute the constraint in every state. For distillation, it would result in the need to have access to the teacher model at test time, which fundamentally defeats the purpose of distillation. To address this issue, we modify the Saute method by removing the state augmentation step using the assumption that the policy is *history-conditioned*, which is the case for LLM. This modification allows us to maintain the **theoretical guarantees** of Saute while ensuring that the student model can operate independently of the teacher at test time. By reformulating the reward function alone, we achieve a more efficient and practical solution for distillation.

Through extensive experiments, we demonstrate that our proposed method effectively minimizes the KL divergence while achieving superior performance in terms of reasoning quality and comparable final answer correctness (see Figure 2). We show that reward maximization alone, as proposed in Guo et al. (2025), cannot guarantee correct reasoning steps by itself and that the teacher signal is useful for LLM to better reason (see Figure 1).

Our contributions are summarized as follows:

- We formulate LLM distillation as a constrained RL problem, integrating task-specific reward signals to guide the distillation process.

- We adapt the Saute method by removing the state augmentation step, ensuring the student model operates independently of the teacher at test time while maintaining the **theoretical guarantees** and **enhancing exploration** on constraint-violating trajectories.

- We conduct extensive experiments on mathematical reasoning tasks to demonstrate that our method identifies a notable point on the Pareto front balancing divergence minimization, reward maximization, and reasoning quality.

This work bridges the gap between distillation and constrained RL, offering a promising direction for improving the efficiency and effectiveness of knowledge transfer in LLMs.

## 2 BACKGROUND

### 2.1 DISTILLATION

Knowledge distillation has emerged as a critical technique for transferring knowledge from large, complex teacher models to smaller, more efficient student models (Hinton et al., 2015). Standard distillation methods primarily focus on minimizing the divergence, often Kullback-Leibler (KL) divergence, between the student and teacher models (Ba & Caruana, 2014; Gou et al., 2021), treating the distillation as a supervised imitation at the token or representation level. While effective for general language understanding tasks, these methods struggle on **complex reasoning tasks**: minimizing solely the divergence while ignoring task-specific reward signals can fail to capture the solution paths with better performance. For instance, in mathematical reasoning tasks, the teacher model may rely on complex reasoning paths that are difficult for a smaller student model to replicate due to its limited capacity, while alternative, simpler reasoning strategies that achieve the same correct outcome might be more suitable for the student to learn and memorize (Zhang et al., 2025).

Recent advances incorporate reward signals into distillation (Agarwal et al., 2024), recasting it as a policy-optimization problem in which the student policy $\pi$ is trained to maximize expected task reward $R$ while being regularized by a divergence $D(\pi, \mu)$ to teacher policy $\mu$:

$$\max_\pi \mathbb{E}_\pi \left[ R - \lambda D(\pi, \mu) \right], \tag{1}$$

where $\lambda$ controls the trade-off between the task performance and teacher fidelity. However, the optimal $\lambda$ is difficult to anticipate and requires extensive retraining on specific tasks, making this approach unstable and computationally expensive for large sequential models. This challenge motivates viewing distillation instead as a **constrained learning problem** that can directly maximize the task reward subject to a divergence budget. This perspective eliminates ad hoc hyperparameter tuning while providing interpretable fidelity guarantees and a principled foundation for reasoning-oriented distillation.

### 2.2 CONSTRAINED REINFORCEMENT LEARNING

Constrained reinforcement learning (CRL) addresses the problem of optimizing a primary objective while satisfying constraint requirements (e.g., safety) (Achiam et al., 2017). In LLM distillation, we can constrain the divergence between the teacher and student policy, following the constrained MDP formulation $\mathcal{M}_d = \langle \mathcal{S}, \mathcal{A}, \mathcal{P}, R, C, \gamma, d \rangle$, where $\mathbf{s}_t$ is the current prompt with partial response, the action $\mathbf{a}_t$ is the next token generated by the student model, $\mathcal{P}$ is the transition kernel, $R$ is the task-specific reward (e.g., correctness in mathematical reasoning), $C_\pi(\mathbf{s_t}) := D_f \left( \pi(\cdot|\mathbf{s}_t) || \mu(\cdot|\mathbf{s}_t) \right)$ is the per-state $f$-divergence between student $\pi$ and teacher $\mu$, $\gamma \in (0, 1)$ is the discount factor and $d$ is a predefined budget. The goal is to find a policy $\pi$ that maximizes the task reward while keeping the expected divergence lower than the threshold $d$:

$$\max_\pi \mathbb{E}_\pi \left[ \sum_{t=0}^\infty \gamma^t R(\mathbf{s}_t, \mathbf{a}_t) \right] \quad \text{s.t.} \quad \mathbb{E}_\pi \left[ \sum_{t=0}^\infty C_\pi(\mathbf{s}_t) \right] \le d. \tag{2}$$

This kind of constrained problem can be solved with a direct optimization: Sootla et al. (2022b) introduced a state augmentation method that reformulates the constrained MDP as an augmented

MDP $\widetilde{\mathcal{M}}_d^n = \langle \tilde{\mathcal{S}}, \mathcal{A}, \tilde{\mathcal{P}}, \tilde{R}_n, \gamma, d \rangle$ by adding a auxiliary state variable $\mathbf{z}_t$ that tracks the remaining budget at every time step $t$, $\mathbf{z}_{t+1} = \mathbf{z}_t - C_\pi(\mathbf{s}_t)$, $\mathbf{z}_0 = d$, transforming the problem into:

$$\max_\pi \mathbb{E}_\pi \left[ \sum_{t=0}^\infty \gamma^t \tilde{R}_n(\mathbf{s}_t, \mathbf{z}_t, \mathbf{a}_t) \right], \quad \tilde{R}_n(\mathbf{s}_T, \mathbf{z}_T, \mathbf{a}_T) = \begin{cases} R(\mathbf{s}_T, \mathbf{a}_T) & \text{if } \mathbf{z}_T \geq 0, \\ -n & \text{if } \mathbf{z}_T < 0, \end{cases} \quad (3)$$

Here $\tilde{\mathcal{S}} = \mathcal{S} \times \mathcal{Z}$ is the augmented state space, $\tilde{\mathcal{P}} : \tilde{\mathcal{S}} \times \mathcal{A} \times \tilde{\mathcal{S}} \to [0, 1]$ is the transition kernel, and $\tilde{R}_n$ is a constrained reward function with a large positive $n \gg R_{\max}$ for penalization when the budget $d$ is exhausted. As $n \to \infty$, any optimal policy of the augmented MDP is feasible for the constraint and attains the constrained optimum under standard assumptions. This method avoids the computational overhead of Lagrange multipliers formulation (cf. equation 1), which can be written as $\max_\pi \min_{\lambda \geq 0} \mathbb{E}_\pi \left[ \sum_{t=0}^\infty \gamma^t R(\mathbf{s}_t, \mathbf{a}_t) - \lambda \left( \sum_{t=0}^\infty C_\pi(\mathbf{s}_t) - d \right) \right]$ for the same formulation, and typically requires tuning a dual variable and running dual ascent.

However, directly applying this formulation to distillation would require maintaining the augmented state $\mathbf{z}_T$ online during distillation, which would necessitate access to the teacher model at test time to compute $C_\pi$ at every timestep. This is impractical for distillation, where the goal is to create a student model that operates independently of the teacher. In the following section, we address this challenge by proposing a new formulation for LLM distillation to eliminate the need for state augmentation while preserving the theoretical guarantees.

## 3 METHOD

### 3.1 CONSTRAINED RL FOR LLM DISTILLATION

We introduce a constrained MDP formulation for distillation that removes state augmentation while retaining the hard-constraint semantics, therefore enabling constrained RL without accessing the teacher policy at every single step. In LLM distillation, we model the state as the full interaction history, so the induced control process is fully observable. Therefore, removing the augmented state $\mathbf{z}_T$ in equation 3 from the state does not induce partial observability. At any time $T$, we can recompute the remaining budget from the full observed history encoded in $\mathbf{s}_T$, hence the augmented state $\mathbf{z}_T$ is a deterministic function of the state with $\mathbf{z}_T = d - \sum_{t=0}^{T-1} C_\pi(\mathbf{s}_t)$.

We propose a constrained MDP formulation for LLM distillation **without** state augmentation $\widehat{\mathcal{M}}_d^n = \langle \mathcal{S}, \mathcal{A}, \mathcal{P}, \hat{R}_n, \gamma, d \rangle$, where $\hat{R}_n$ is the constrained reward that combines the task-specific reward with a feasibility signal for constraint satisfaction. The goal is to find a policy $\pi$ that maximizes the task-specific reward while keeping the divergence lower than the threshold $d$:

$$\max_\pi \mathbb{E}_\pi \left[ \sum_{t=0}^\infty \gamma^t \hat{R}_{\pi,n}(\mathbf{s}_t, \mathbf{a}_t) \right], \quad \hat{R}_{\pi,n}(\mathbf{s}_T, \mathbf{a}_T) = \begin{cases} R(\mathbf{s}_T, \mathbf{a}_T) & \text{if } d - \sum_{t=0}^{T-1} C_\pi(\mathbf{s}_t) \geq 0, \\ -(n+\phi_\pi(\mathbf{s_T})) & \text{otherwise.} \end{cases}$$

$$(4)$$

The constrained reward **without the augmented state** $\mathbf{z}_T$ preserves the feasibility signal for the constraint satisfaction, such that the student model receives the positive task-specific reward only when the constraint with budget $d$ is satisfied, while any trajectory that violates the constraint incurs a large hard penalty. This penalty is a fixed value in the previous setting equation 3 for all infeasible trajectories, we refine this penalty by adding **a policy-dependent discrepancy term** $\phi_\pi(\mathbf{s}_T)$ to differentiate the trajectories within the infeasible region: trajectories that deviate more from the teacher policy receive a stronger penalty, whereas marginally deviating ones are penalized less. We define $\phi_\pi(\mathbf{s}_T)$ as any $f$-divergence, including KL and Jensen–Shannon divergence, between the student and teacher at $\mathbf{s}_T$, which is nonnegative and equals zero iff $\pi(\cdot \mid \mathbf{s}_T) = \mu(\cdot \mid \mathbf{s}_T)$. Therefore, the penalty $-(n + \phi_\pi(\mathbf{s}_T))$ is strictly negative, while in feasible region we maintain the original task specific reward to guide exploration. This formulation preserves the augmented-MDP penalty semantics and increases sample efficiency by delivering informative negative feedback among violating trajectories, without altering feasibility decisions or the limiting optimum.

### 3.2 Policy Gradient Optimization

We detail policy gradient optimization for the unaugmented objective in equation 4, and derive the policy gradient decomposition with an explicit-dependence term. Our method directly maximizes the expected discounted return with standard policy gradient, thereby avoiding the instabilities from infeasible gradient vector fields in on-policy distillation observed by Czarnecki et al. (2019). We parameterize the student policy as $\pi_\theta$ and maximize the expected discounted return:

$$J_n(\theta) = \mathbb{E}_{\pi_\theta}\Big[ \sum_{t=0}^{\infty} \gamma^t \hat{R}_{\pi_\theta,n}(\mathbf{s}_t, \mathbf{a}_t)\Big]$$

Because $J_n(\theta)$ depends on $\theta$ both through the trajectory distribution induced by $\pi_\theta$ and inside the reward via the discrepancy $\phi_{\pi_\theta}$, its gradient decomposes into (I) the likelihood-ratio term and (II) the explicit dependence term of $\hat{R}_{\pi_\theta,n}$ on $\theta$:

$$\nabla_\theta J_n(\theta) = \underbrace{\mathbb{E}_{\pi_\theta}\left[ \sum_{t\geq 0} \nabla_\theta \log \pi_\theta(\mathbf{a}_t \mid \mathbf{s}_t) \left( \sum_{u\geq t} \gamma^{u-t} \hat{R}_{\pi_\theta,n}(\mathbf{s}_u, \mathbf{a}_u)\right)\right]}_{\text{(I) likelihood-ratio term}} + \underbrace{\mathbb{E}_{\pi_\theta}\left[ \sum_{t\geq 0} \gamma^t \, \partial_\theta \hat{R}_{\pi_\theta,n}(\mathbf{s}_t, \mathbf{a}_t)\right]}_{\text{(II) explicit-dependence term}}$$

$$(5)$$

We compute $\nabla_\theta J_n(\theta)$ following the policy gradient theorem Sutton et al. (1999) under the following minimal assumptions:

**A1.** For each state $\mathbf{s}_T$, $\phi_{\pi_\theta}(\mathbf{s}_T)$ is finite and differentiable in $\theta$, and its gradient is measurable and integrable along trajectories $\mathbb{E}_{\pi_\theta}\big[ \sum_{t\geq 0} \gamma^t \|\partial_\theta \phi_{\pi_\theta}(\mathbf{s}_t)\|\big] < \infty$;

**A2.** There exists an optimal policy $\pi_\theta^*$ with a finite value such that $\mathbb{P}\Big(d - \sum_{t=0}^{T-1} C_{\pi_\theta^*}(\mathbf{s}_t) > 0\Big) = 1$.

In practice, we take $\phi = \mathrm{KL}$ with a small probability floor, ensuring finiteness and differentiability. A2 ensures the existence of an optimal feasible policy, i.e., the budget is satisfied almost surely at the optimum. Under A1 and A2, we can characterize the explicit-dependence term (II) in a unified way (see Appendix A for the full derivation across feasible, infeasible, and boundary cases) by including the gradient and limiting sub-gradient of $\hat{R}_{\pi_\theta,n}$ with a small tolerance $\varepsilon \downarrow 0$ round the feasibility boundary. Our final gradient for optimization is

$$\nabla_\theta J_n(\theta) = \mathbb{E}_{\pi_\theta}\left[ \sum_{t\geq 0} \nabla_\theta \log \pi_\theta(\mathbf{a}_t \mid \mathbf{s}_t) \left( \sum_{u\geq t} \gamma^{u-t} \hat{R}_{\pi_\theta,n}(\mathbf{s}_u, \mathbf{a}_u)\right)\right]$$

$$- \mathbb{E}_{\pi_\theta}\left[ \sum_{t\geq 0} \gamma^t \, \mathbb{1}\left\{ d - \sum_{u=0}^{t-1} C_{\pi_\theta}(\mathbf{s}_u) \leq \varepsilon \right\} \partial_\theta \phi_{\pi_\theta}(\mathbf{s}_t)\right]$$

### 3.3 Theoretical Guarantee for Constraint Satisfaction

In this section, we show that our reformulation of the constrained MDP preserves the constraint satisfaction guarantee while enabling deployment without teacher access. In particular: (i) the optimal policy and value functions are equivalent between our un-augmented objective in equation 4 and the augmented objective in equation 3; (ii) Bellman optimality holds under standard assumptions; and (iii) as $n \to \infty$, every optimal policy with finite value satisfies the constraint almost surely.

In LLM distillation, the student policy $\pi$ is frozen within each episode, so the induced control process is time-homogeneous. We adopt this per-episode stationary view; all statements are uniform over a fixed $\pi$ on the reachable set. We further formalize an equivalent *contextual MDP* view, in which each episode carries a fixed context $c$ (e.g., a policy checkpoint), and prove its optimality-equivalence to the standard MDP in Appendix C.

**Theorem 3.1** (Optimal equivalence). *For every feasible state $\mathbf{s}_T$, the optimal value functions of the unaugmented MDP $\widehat{\mathcal{M}}_d^n$ in equation 4 and the augmented MDP $\widetilde{\mathcal{M}}_d^n$ in equation 3 are equivalent:*

$$\hat{V}^*(\mathbf{s}_T) = \tilde{V}^*(\mathbf{s}_T, \mathbf{z}_T).$$

This theorem justifies that removing the budget variable $\mathbf{z}_t$ does not change the control problem we are solving. This equivalence holds because the augmented state $z_T$ is *reconstructable* from the observed history under any fixed student $\pi$ and teacher $\mu$ via $\mathbf{z}_T = d - \sum_{t=0}^{T-1} C_\pi(\mathbf{s}_t)$, so augmented states $(\mathbf{s}_T, \mathbf{z}_T)$ and un-augmented states $\mathbf{s}_T$ induce identical trajectories and stepwise rewards along any feasible paths. We give the precise construction and full proof details in Appendix B.

We adopt the following standard assumptions Hernández-Lerma & Muñoz de Ozak (1992); Sootla et al. (2022b) for the discrete token setting in distillation:

**B1.** The reward function $\hat{R}_n(\mathbf{s}_T, \mathbf{a}_T)$ is bounded, measurable, and upper semicontinuous on $\mathcal{S} \times \mathcal{A}$;

**B2.** The transition kernel $\mathcal{P}$ is weakly continuous on $\mathcal{S} \times \mathcal{A}$;    **B3.** The action space $\mathcal{A}$ is compact.

**Theorem 3.2** (Bellman optimality and value convergence). *Consider the unaugmented MDP $\widehat{\mathcal{M}}_d$, satisfying assumption B1-B3 with the associated equation 4, then:*

*a) the Bellman equation is satisfied in $\widehat{\mathcal{M}}_d$;*

*b) the optimal value function $\hat{V}_n^*$ for $\widehat{\mathcal{M}}_d^n$ converges monotonically to $\hat{V}_\infty^*$ for $\widehat{\mathcal{M}}_d^\infty$.*

**Theorem 3.3** (Almost surely constraint satisfaction). *If there exists an optimal policy $\pi^*$ solving $\widehat{\mathcal{M}}_d^\infty$ with a finite value, then $\pi^*$ is also an optimal policy for the original constrained MDP $\mathcal{M}_d$ and satisfies the constraint almost surely.*

These results show that our modified approach maintains the guarantees of the original constrained problem while eliminating state augmentation (see Appendix B for proofs and discussion). At test time, the student operates without teacher access: the cumulative reward is computed from the student's own output distribution and environment feedback. This makes our approach practical for LLM distillation while retaining guarantees of feasibility.

## 4 EXPERIMENTS

### 4.1 EXPERIMENTAL SETUP

We conduct experiments on two distinct distillation settings to evaluate our proposed method. For the first setting, we distill a *Qwen2.5-1.5B-Math* student model from a *Qwen2.5-7B-Math-Instruct* teacher model using the GSM8K training dataset. For the second setting, we distilled a *Llama-3.2-3B* student model from a *Llama-3.2-11B-Instruct* teacher model using the MATH training dataset. In both setting, we evaluated the resulting checkpoints after 20 epochs on the Apple/GSM-Symbolic (main) (Mirzadeh et al., 2025), the test set of GSM8K (Cobbe et al., 2021) and the whole test set of MATH (Hendrycks et al., 2021) (from which MATH500 is selected).

**Baselines.** Our proposed constrained optimization method is built upon the GRPO policy gradient algorithm (Shao et al., 2024). To assess its effectiveness, we benchmark against several strong distillation baselines, each re-implemented under the same GRPO framework to ensure fairness and consistency. More precisely, for every method, the batch size and its composition is the same (64 answers, 8 questions, 8 answers per question). The learning rate ($1e^{-5}$) and the optimizer (AdamW) are also the same. We consider the following baselines:

- **GRPO**: The base algorithm in our experiments. GRPO optimizes purely for the *task-specific reward* using a robust, value-function-free policy gradient with a group-average reward baseline (Shao et al., 2024).

- **GKD**: A distillation-only baseline whose objective is to minimize the *reverse* KL divergence $D_{\mathrm{KL}}(\pi_\theta \parallel \mu)$, treating the negative per-step KL as an intrinsic reward. We use GRPO rather than the REINFORCE-style update of Agarwal et al. (2024) for consistency.

- **GKD-GRPO**: A baseline that jointly optimizes for both the task-specific reward and the GKD objective. This corresponds to the standard Lagrangian relaxation of our constrained problem in Eq. (1), with $\lambda$ as the balancing hyperparameter (Agarwal et al., 2024).

- **Mini-LLM**: On-policy *reverse* KL divergence minimization (Gu et al., 2024), accounting for the long-term effects of actions on KL (Tang & Munos, 2025). As in GKD, task reward is ignored.

For consistency, we sample trajectories exclusively with the student policy and substitute PPO with a GRPO-based update.

Together, these baselines span the main approaches to RL-based distillation: optimizing task rewards, relying solely on KL supervision, and hybrid formulations that combine both. To approximate the Pareto frontier of the Lagrangian relaxation baseline (GKD-GRPO), we perform a grid search over the multiplier $\lambda$ across several orders of magnitude, reporting results for $\lambda \in \{0.001, 0.01, 0.1, 1.0, 10\}$. Note that when $\lambda = 0$, it equals to the pure GRPO baseline. The constraint threshold $d = 0.35$ was selected based on preliminary experiments that seek to minimize only the KL (mini-LLM and GKD).

**Metrics.** We evaluate models using four key metrics:

- **Final Answer Correctness** (FAC): It verifies that the final answer inside \boxed{} is correct. It is used to define the reward function $R$ in our MDPs.

- **Reasoning Quality**: To assess the logical validity of the reasoning path beyond the final answer, we use an LLM-as-a-Judge setting (Zheng et al., 2023). Specifically, we use *DeepSeek-R1-Distill-Qwen-32B* (DeepSeek-AI, 2025) to perform pairwise comparisons between generated solutions. The judge is provided with the correct final answer to isolate its evaluation to the reasoning process itself. This yields the *Reasoning Win Rate* (**RWR**) and *Reasoning Loss Rate* (**RLR**), reported as percentages (Zhou et al., 2025).

- **Constraint Satisfaction**: The percentage of test samples where the KL divergence between the student and teacher policies is below a predefined threshold $d$.

- **KL Divergence**: The average student-teacher policy divergence cross the entire test set.

### 4.2 EXPERIMENT RESULTS

We organize our set of experiments to answer the following questions:

- A. What is the best method in general?
- B. Is our method able to achieve higher constraints satisfaction?
- C. Can external reward help achieve better distillation?
- D. Does the distillation signal help to better reason?

**A. What is the best method in general?** Figure 2 presents a comprehensive comparison of our constrained RL approach against baseline methods across five key metrics. The results demonstrate that our method achieves the most balanced performance profile, excelling particularly in reasoning quality and constraint satisfaction while maintaining competitive final answer correctness. The radar plot reveals that pure reward optimization (GRPO $\lambda$=0.0) achieves the highest final answer correctness but at the cost of poor reasoning quality and severe constraint violations. Conversely, methods that focus solely on KL minimization (GKD, Mini-LLM) maintain good constraint satisfaction but suffer from lower final answer correctness. Our constrained RL formulation successfully navigates this trade-off, achieving strong performance across all dimensions.

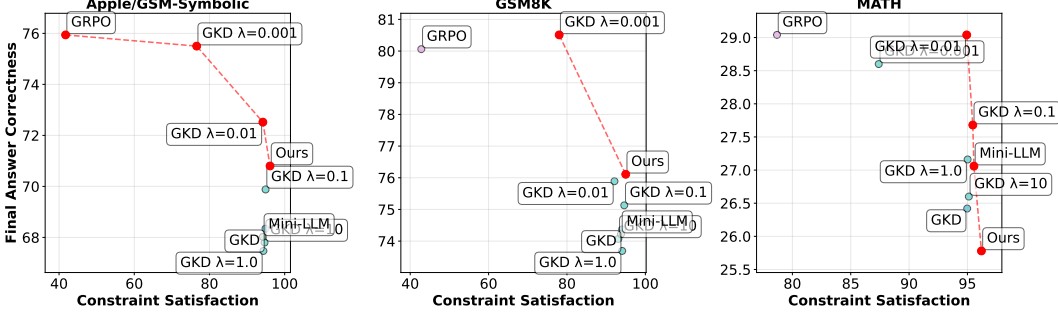

Figure 3: Pareto frontier analysis showing the trade-off between final answer correctness and constraint satisfaction across different methods and hyperparameter settings for Qwen2.5-1.5B-Math. Each point represents a different method configuration. The points in red belong to the Pareto front.

**B. Is our method able to achieve higher constraints satisfaction?** Figure 3 illustrates the Pareto frontier between final answer correctness and constraint satisfaction across different methods and hyperparameter settings. Our approach consistently achieves superior constraint satisfaction rates while maintaining competitive final answer correctness, occupying a unique region of the Pareto front. This demonstrates the effectiveness of our constrained formulation in achieving the desired balance between task performance and teacher fidelity. Note that without introducing $\phi$, our method would have a great difficulty satisfying a strict constraint due to the lack of signal: all trajectories would receive the same penalty $n$ and the training would divergence.

**C. Can external reward help achieve better distillation?** Comparing reward-based methods (GRPO, GKD-GRPO variants, and ours) against purely KL-based methods (GKD, Mini-LLM) reveals the crucial role of external rewards in distillation. Pure KL minimization methods always achieve lower final answer correctness rates in every dataset for each model (Figure 2 and Appendix E). Beyond the improvement over final answer correctness, we also observe that our method achieves higher reasoning win rates which can also be attributed to the use of the reward function. This substantial improvement demonstrates that incorporating task-specific rewards enables the student model to learn more effective reasoning strategies rather than merely mimicking the teacher's surface-level outputs.

**D. Does the distillation signal help to better reason?** Figure 4 presents a comprehensive pairwise comparison matrix averaged across all three evaluation datasets with Qwen. The comparison between pure reward optimization (GRPO $\lambda$=0.0) and our constrained approach provides strong evidence for the value of teacher guidance in reasoning tasks. While GRPO achieves the highest raw final answer correctness (75-80%), it exhibits poor reasoning quality with win rates of only 12-19% and correspondingly high loss rates of 39-55%. Our constrained formulation dramatically improves reasoning quality while maintaining competitive success rates. It demonstrates that constraining the student to stay close to the teacher distribution helps preserve and transfer the teacher's reasoning capabilities. The equivalent figure for Llama3.2-3B is provided in the Appendix F.

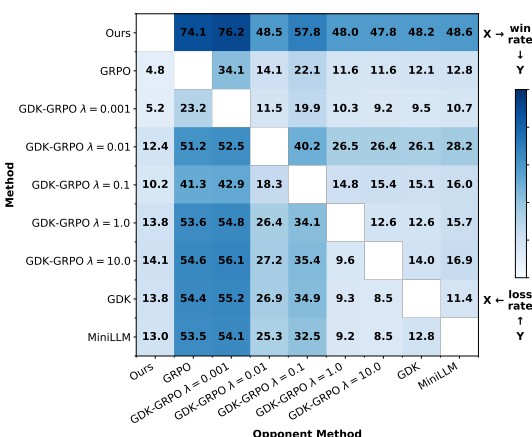

Figure 4: Pairwise comparison heatmap showing the relative performance of our method against baselines, averaged across all three evaluation datasets (Apple/GSM-Symbolic, GSM8K, and MATH) with Qwen2.5-3B-Math. Darker colors indicate superior performance in row-to-column comparisons. Averaging over columns gives the reasoning win rate (RWR) and over rows the reasoning loss rate (RLR).

*Qualitative Analysis*: In Figure 1, we present a test set example in which both our method and the GRPO baseline yield the correct final answer. However, only our method produces logically valid reasoning steps, while GRPO's reasoning is flawed. More examples are provided in the Appendix F.

These results collectively demonstrate that our constrained RL approach successfully addresses the core challenges of reward-aware distillation: it maintains high constraint satisfaction rates, leverages external rewards for improved task performance, and preserves the teacher's reasoning capabilities in the student model. The method achieves a superior balance across all evaluation dimensions compared to existing approaches that typically excel in only one aspect of the distillation objective. We provide more detailed results in the Appendix E.

## 5 RELATED WORKS

**Task-specific Distillation.** The prevailing paradigm in LLM distillation is to pass knowledge from a powerful teacher to a compact student by aligning their output distributions, typically through the

*reverse* KL divergence (Hinton et al., 2015; Sanh et al., 2020; Gu et al., 2024; Agarwal et al., 2024). However, this objective does not explicitly guarantee the preservation of the teacher's underlying reasoning abilities on complex tasks (Gudibande et al., 2024), motivating a shift towards more sophisticated, task-aware techniques. The problem is now increasingly framed through the lens of RL, where adherence to the teacher is elegantly re-conceptualized as a dense, token-level reward derived from the KL divergence. This forms the basis of general-purpose distillation methods (Agarwal et al., 2024; Ko et al., 2024; 2025), which uses a REINFORCE-style update, and Mini-LLM (Gu et al., 2024), which decomposes the policy gradient to separate the high-variance, long-term reward from a more stable, single-step objective. This RL framework can then be extended by composing the KL-based reward with an external task reward, $R_{\text{task}}$ (Agarwal et al., 2024).

**Task-aware Extensions.** Beyond these RL formulations, a significant body of work integrates richer, task-specific signals into the distillation process to provide denser supervision. One prominent strategy, *process-aware* distillation, supervises the student to replicate the teacher's intermediate reasoning steps, thereby transferring the underlying causal logic rather than just the final output (Hsieh et al., 2023; Adarsh et al., 2024; Chen et al., 2025). Other approaches include *logit-aware* distillation, which intelligently modifies the KL divergence loss to emphasize pivotal, task-relevant tokens identified via attention or Bayesian principles (Li et al., 2025; 2024; Saadi & Wang, 2025), and *knowledge-augmented* methods that use retrieval to transfer a teacher's ability to synthesize external information (Kang et al., 2023; Tian et al., 2025). While these sophisticated strategies significantly improve signal density, they often introduce new complexities, such as the need for fine-grained annotations, complex weighting heuristics, or the overhead of external knowledge bases.

**Constrained RL for LLM Distillation.** The application of RL to task-specific LLM distillation remains relatively under-explored (Zhang et al., 2025). In standard alignment settings like RLHF (Ouyang et al., 2022), the KL penalty against a reference model is primarily a regularization tool to prevent catastrophic forgetting and maintain stylistic diversity (Yang et al., 2024; Stiennon et al., 2022). However, in the distillation setting, this KL term takes on the dual role of a constraint, intended to preserve the teacher's reasoning capabilities. Most methods still use a fixed penalty, which is simple but can be brittle, as a static weight may not prevent the student from exploiting task rewards via shallow or degenerate reasoning (Gudibande et al., 2024). To our knowledge, the principled distillation of task-specific, constrained RL policies from LLMs is still scarce, with most related work only examining it briefly (Agarwal et al., 2024).

A more robust alternative is to treat the KL divergence as an explicit trust-region constraint and solve the resulting constrained-RL problem; classic trust-region and constrained-RL methods provide a standard toolkit for this (Schulman et al., 2015; Achiam et al., 2017). Dual Lagrangian solvers can then adapt the KL penalty to restore an interpretable fidelity–performance point, but at LLM scale, this is practically challenging: teacher forward passes, cached-logit strategies, and inner-loop/dual updates add significant compute, memory, and variance costs (Dasgupta et al., 2023; Achiam et al., 2017). In this work, we address these challenges by reformulating the dual Lagrange problem within a *state-augmented* MDP framework (Calvo-Fullana et al., 2024; Sootla et al., 2022a;b), for which we provide a principled and efficient optimization solution that remains practical at the LLM scale.

## 6  CONCLUSION

In this work, we moved beyond the conventional paradigm of regularized distillation and introduced a principled framework based on constrained reinforcement learning. By adapting principles from the safe RL literature, we developed a solution that maintains theoretical guarantees of constraint satisfaction without requiring the impractical state augmentation typical of classic methods. This approach successfully navigates the trade-off between task-specific performance and teacher fidelity, eliminating the need for brittle, ad-hoc reward weighting and the prohibitive costs of traditional dual max-min optimization. Our experiments on mathematical reasoning demonstrate that it is possible to enforce a strict KL divergence constraint with high fidelity while maintaining competitive task rewards. This method provides a theoretically grounded and practically efficient pathway for creating smaller, reliable, and specialized models that operate reliably within a defined trust region of their teacher—a crucial step towards more controllable and deployable LLMs.

**Ethics Statement** We adhere to the ICLR Code of Ethics, and this work does not involve any potential ethical concerns or violations.

**Reproducibility statement** We provide full training details and hyperparameter settings, complete proofs, along with citations for the open-source base model and dataset, in the main text and appendix. We will release our source code upon acceptance.

**The Use of Large Language Models** LLMs were employed as auxiliary tools during the manuscript preparation process to enhance the clarity and conciseness of written content. Specifically, LLMs were utilized for linguistic refinement tasks, including sentence restructuring, word choice optimization, and text compression to improve overall readability while maintaining the original meaning and scientific accuracy of the content.

It is important to note that all LLM-suggested modifications were carefully reviewed and validated by the authors before incorporation into the final text. The authors assume full responsibility for the accuracy and validity of all content presented in this work.

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

# A DERIVATION OF POLICY GRADIENT

We compute the gradient of $J_n(\theta)$ w.r.t $\theta$ following the policy gradient theorem Sutton et al. (1999) under the following minimal assumptions:

**A1.** For each state $\mathbf{s}$, $\phi_{\pi_\theta}(\mathbf{s})$ is finite and differentiable in $\theta$, and its gradient is measurable and integrable along trajectories: $\mathbb{E}_{\pi_\theta}\big[\sum_{t\geq 0}\gamma^t\|\partial_\theta\phi_{\pi_\theta}(\mathbf{s}_t)\|\big] < \infty$;

**A2.** There exists an optimal policy $\pi_\theta^*$ with a finite value such that $\mathbb{P}\Big(d - \sum_{t=0}^{T-1}C_{\pi_\theta^*}(\mathbf{s}_t) > 0\Big) = 1$.

Assumption A1 ensures that the discrepancy function $\phi_{\pi_\theta}$ and its gradient are well-behaved so that the explicit-dependence term (II) in equation 5 is finite and integrable to guarantee that the policy-gradient estimator has bounded variance. This assumption can be satisfied by many discrepancy functions, in our implementation, we choose $\phi$ as the KL divergence $\phi_{\pi_\theta}(\mathbf{s}) = \mathrm{KL}(\pi_\theta(\cdot \mid \mathbf{s})\|\mu(\cdot \mid \mathbf{s}))$, whose gradient admits the standard score-function $\partial_\theta\phi_{\pi_\theta}(\mathbf{s}) = \mathbb{E}_{a\sim\pi_\theta(\cdot\mid\mathbf{s})}\Big[\nabla_\theta\log\pi_\theta(a \mid \mathbf{s})\big(1 + \log\pi_\theta(a \mid \mathbf{s}) - \log\mu(a \mid \mathbf{s})\big)\Big]$. By enforcing overlapping support between $\pi_\theta$ and $\mu$ in implementation (e.g., using a probability floor), we guarantee that $\phi_{\pi_\theta}$ remains finite and that $\partial_\theta\phi_{\pi_\theta}$ is bounded across all states, thereby satisfying assumption A1.

Assumption A2 requires that the optimal policy $\pi_\theta^*$ exists inside the feasible set, which implies that the budget constraint is almost surely satisfied and no probability mass is concentrated on the boundary. This assumption is mild in practice, since by choosing a sufficiently large penalty parameter $n$ we can always discourage boundary-violating policies and guarantee the existence of a feasible optimum.

Under assumptions A1–A2, we can characterize the explicit-dependence term (II) in a unified way:

1) On strictly feasible trajectories, i.e., when $d - \sum_{u=0}^{t-1}C_{\pi_\theta}(\mathbf{s}_u) > 0$, the feasibility indicator is locally constant in a neighborhood of $\pi_\theta^*$, so $\partial_\theta\hat{R}_{\pi_\theta,n}(\mathbf{s}_t,\mathbf{a}_t) = 0$ at every step and term (II) vanishes.

2) When a trajectory has already violated the budget, the reward switches to the penalized branch, therefore in the infeasible region term (II) reduces to $\partial_\theta\hat{R}_{\pi_\theta,n}(\mathbf{s}_t,\mathbf{a}_t) = -\partial_\theta\phi_{\pi_\theta}(\mathbf{s}_t)$.

3) At the boundary, where the cumulative constraint exactly equals $d$, the reward becomes non-differentiable. We replace the derivative with a generalized subgradient, following prior RL works with non-smooth objectives Zhang et al. (2020); Wang & Zou (2022); Kumar et al. (2023). We adopt the Mordukhovich subgradient following the definition from Mordukhovich (2018), and the term (II) reduces to $-\mathbb{1}\big\{d - \sum_{u=0}^{t-1}C_{\pi_\theta}(\mathbf{s}_u) \leq \varepsilon\big\}\partial_\theta\phi_{\pi_\theta}(\mathbf{s}_t)$ by taking the limiting subgradient from the infeasible side with a small tolerance $\varepsilon \downarrow 0$ during training.

We note that in practice, the probability of hitting the boundary exactly is small in the continuous setting of the constraint value, and term (II), through its explicit single-step decomposition, also contributes to variance reduction during training, as observed in prior works Czarnecki et al. (2019); Gu et al. (2024). As a result, term (II) disappears on feasible trajectories near the optimum, while continuing to provide informative signals both for trajectories that violate the constraint and for those approaching the boundary.

Therefore, our final gradient for optimization is

$$\nabla_\theta J_n(\theta) = \mathbb{E}_{\pi_\theta}\left[\sum_{t\geq 0}\nabla_\theta\log\pi_\theta(\mathbf{a}_t \mid \mathbf{s}_t)\left(\sum_{u\geq t}\gamma^{u-t}\hat{R}_{\pi_\theta,n}(\mathbf{s}_u,\mathbf{a}_u)\right)\right]$$

$$- \mathbb{E}_{\pi_\theta}\left[\sum_{t\geq 0}\gamma^t\,\mathbb{1}\left\{d - \sum_{u=0}^{t-1}C_{\pi_\theta}(\mathbf{s}_u) \leq \varepsilon\right\}\partial_\theta\phi_{\pi_\theta}(\mathbf{s}_t)\right]$$

# B  PROOFS OF CONSTRAINT SATISFACTION GUARANTEE

**Theorem B.1** (Optimal equivalence). *For every feasible state $\mathbf{s}_T$, the optimal value functions of the unaugmented MDP $\widehat{\mathcal{M}}_d^n$ in equation 4 and the augmented MDP $\widetilde{\mathcal{M}}_d^n$ in equation 3 are equivalent:*

$$\hat{V}^*(\mathbf{s}_T) = \tilde{V}^*(\mathbf{s}_T, \mathbf{z}_T).$$

*Proof.* Given the budget recursion $\mathbf{z}_T = d - \sum_{t=0}^{T-1} C_\pi(\mathbf{s}_t)$ and the fact that $\mathbf{s}_T$ encodes the whole past and $C_\pi$ is deterministic in $\mathbf{s}$ given a fixed teacher policy $\mu$ and a student policy $\pi$, $\mathbf{z}_T$ is a deterministic function of any reachable $\mathbf{s}_T$ for any predefined budget $d$. Therefore, the step-wise rewards in the feasible set are equivalent in $\widetilde{\mathcal{M}}_d^n$ and $\widehat{\mathcal{M}}_d^n$, $\tilde{R}_n(\mathbf{s}_T, \mathbf{z}_T, \mathbf{a}_T) = \hat{R}_{\pi,n}(\mathbf{s}_T, \mathbf{a}_T)$ for every reachable time $T$ along any feasible trajectories by the definitions in equation 3 and equation 4.

The $\mathbf{s}$-marginal transition kernel is identical in both formulations $\mathbf{s}_{T+1} \sim \mathcal{P}_\mathcal{S}(\cdot \mid \mathbf{s}_T, \mathbf{a}_T)$, and the budget update is deterministic $\mathbf{z}_{T+1} = \mathbf{z}_T - C_\pi(\mathbf{s}_T)$ in the augmented model $\widetilde{\mathcal{M}}_d^n$. Define the projected policy on the reachable set by $\bar{\pi}(\mathbf{a} \mid \mathbf{s}) := \pi(\mathbf{a} \mid \mathbf{s}, \mathbf{z}(\mathbf{s}))$, where $\mathbf{z}(\mathbf{s})$ denotes the reconstructed budget associated with $\mathbf{s}$. Then the action distribution under $\bar{\pi}$ at $\mathbf{s}$ equals that under $\pi$ at $(\mathbf{s}, \mathbf{z}(\mathbf{s}))$. Therefore, the induced $(\mathbf{s}, \mathbf{a})$-trajectory laws coincide, and together with the step-wise reward equality we obtain the policy-wise identity $\hat{V}_n^{\bar{\pi}}(\mathbf{s}_T) = \tilde{V}_n^\pi(\mathbf{s}_T, \mathbf{z}_T)$.

Conversely, for any un-augmented policy $\bar{\pi}(\mathbf{a} \mid \mathbf{s})$ define the lifted policy $\pi^\uparrow(\mathbf{a} \mid \mathbf{s}, \mathbf{z}) := \bar{\pi}(\mathbf{a} \mid \mathbf{s})$. This yields $\tilde{V}_n^{\pi^\uparrow}(\mathbf{s}, \mathbf{z}) = \hat{V}_n^{\bar{\pi}}(\mathbf{s})$ on the reachable set, so the suprema over the two policy classes agree there; hence $\hat{V}_n^*(\mathbf{s}_T) = V_n^*(\mathbf{s}_T, \mathbf{z}_T)$. $\square$

We adopt the following standard assumptions Hernández-Lerma & Muñoz de Ozak (1992); Sootla et al. (2022b) for the discrete token setting in distillation:

**B1.** The reward function $\hat{R}_n(\mathbf{s}_T, \mathbf{a}_T)$ is bounded, measurable, and upper semicontinuous on $\mathcal{S} \times \mathcal{A}$;

**B2.** The transition kernel $\mathcal{P}$ is weakly continuous on $\mathcal{S} \times \mathcal{A}$;    **B3.** The action space $\mathcal{A}$ is compact.

**Theorem B.2** (Bellman optimality and value convergence). *Consider the unaugmented MDP $\widehat{\mathcal{M}}_d$, satisfying assumption B1-B3 with the associated equation 4, then:*

*a) the Bellman equation is satisfied in $\widehat{\mathcal{M}}_d$;*

*b) the optimal value function $\hat{V}_n^*$ for $\widehat{\mathcal{M}}_d^n$ converges monotonically to $\hat{V}_\infty^*$ for $\widehat{\mathcal{M}}_d^\infty$.*

*Proof.* For **B1**, the task reward in our setting is bounded and measurable on feasible steps, $0 \le R(\mathbf{s}, \mathbf{a}) \le R_{\max}$, and the discrepancy on infeasible steps is also bounded and measurable, $0 \le \phi_\pi(\mathbf{s}) \le \Phi_{\max}$. On the discrete token state–action space $(\mathcal{S} \times \mathcal{A})$, every real-valued function is continuous and hence also upper semicontinuous. Since each point is isolated, any sequence $(\mathbf{s}_k, \mathbf{a}_k) \to (\mathbf{s}, \mathbf{a})$ is eventually constant, so $\limsup_{(\mathbf{s}', \mathbf{a}') \to (\mathbf{s}, \mathbf{a})} \hat{R}_n(\mathbf{s}', \mathbf{a}') = \hat{R}_n(\mathbf{s}, \mathbf{a})$, which establishes B1.

For **B2**, note that for any bounded function $g : \mathcal{S} \to \mathbb{R}$, the map $(\mathbf{s}, \mathbf{a}) \mapsto \sum_{\mathbf{s}'} \mathcal{P}(\mathbf{s}' \mid \mathbf{s}, \mathbf{a}) g(\mathbf{s}')$ is continuous since the domain is discrete, which implies the usual weak continuity condition holds in this setting.

For **B3**, the action set $\mathcal{A}$ is a finite token space, hence compact.

a) Under B1–B3, standard dynamic programming results ensure the existence of an optimal value function satisfying the Bellman equation for $\widehat{\mathcal{M}}_d^n$ by using Theorem 4.2 in Hernández-Lerma & Muñoz de Ozak (1992), applied here to the discrete setting.

b) The penalty on infeasible steps becomes harsher with $n$ while using the same discrepancy function $\phi_\pi$. Let $m > n$, then on infeasible steps $\hat{R}_m \le \hat{R}_n$. Hence $\hat{V}_m^\pi(\mathbf{s}) \le \hat{V}_n^\pi(\mathbf{s})$ for any policy $\pi$ and state $\mathbf{s}$, and taking $\sup_\pi$ yields $\hat{V}_m^*(\mathbf{s}) \le \hat{V}_n^*(\mathbf{s})$. Therefore, the optimal values $\hat{V}_n^*$ converge monotonically to $\hat{V}_\infty^*$ as $n \to \infty$. $\square$

**Theorem B.3** (Almost surely constraint satisfaction). *If there exists an optimal policy $\pi^*$ solving $\widehat{\mathcal{M}}_d^\infty$ with a finite value, then $\pi^*$ is also an optimal policy for the original constrained MDP $\mathcal{M}_d$ and satisfies the constraint almost surely.*

*Proof.* In $\widehat{\mathcal{M}}_d^\infty$, any trajectory that ever violates the budget receives $-\infty$ return; therefore a finite value under $\pi^*$ implies $\mathbb{P}_{\pi^*}(\sum_{t=0}^\infty C_{\pi^*}(\mathbf{s}_t) \leq d) = 1$, i.e., the constraint holds almost surely. On the feasible set, where the budget is never violated, the step-wise rewards in $\widehat{\mathcal{M}}_d^\infty$ and $\mathcal{M}_d$ coincide, so the objectives coincide. Since $\pi^*$ maximizes the objective in $\widehat{\mathcal{M}}_d^\infty$ and is feasible almost surely, it also maximizes the objective in $\mathcal{M}_d$ and satisfies the constraint almost surely. □

## C    A PERSPECTIVE OF LLM DISTILLATION AS CONTEXTUAL MDPs

We formalized LLM distillation as a standard MDP in this work, given that the student $\pi_\theta$ is frozen within each episode and the teacher $\mu$ is fixed during distillation, so the induced control process is time-homogeneous. This is the standard formulation used in prior RL for LLM distillation works Gu et al. (2024); Czarnecki et al. (2019) and supports standard convergence/optimality analysis. Here we note an equivalent viewpoint that treats each episode under a fixed *context* $c$ (e.g., a policy check-point), giving a *Contextual MDP* that is optimality equivalent to the standard MDP formulation.

**Definition C.1** (Contextual MDP for LLM Distillation). The contextual MDP $\mathcal{M}_d^{\text{ctx}}$ is a tuple $(\mathcal{C}, \mathcal{S}, \mathcal{A}, P, R_n^{\text{ctx}}, \gamma)$, where $\mathcal{C}$ is the context space, with $c \in \mathcal{C}$ fixed during an episode, the contextual reward $R_n^{\text{ctx}} : \mathcal{S} \times \mathcal{A} \times \mathcal{C} \to \mathbb{R}$ is

$$
R_n^{\text{ctx}}(s, a; c) = \begin{cases} R(s, a), & \text{if } d - \sum_{t=0}^{T-1} C(s_t, c) \geq 0, \\[2mm] -\big(n + \phi(s, c)\big), & \text{otherwise.} \end{cases}
$$

with $C(\cdot, c)$ the per-step constraint at context $c$ and $\phi(s, c)$ any f-divergence (e.g., $\phi(s, c) = \text{KL}\big(\pi_c(\cdot \mid s) \| \mu(\cdot \mid s)\big)$). A *contextual policy* is a Markov kernel $\pi(\cdot \mid s, c)$ on $\mathcal{A}$.

For any fixed $c$, the slice of $\mathcal{M}_d^{\text{ctx}}$ at that context induces the per-episode stationary problem used in $\widehat{\mathcal{M}}_d$, with per-context reward $\hat{R}_{\pi_c, n}(s, a) := R_n^{\text{ctx}}(s, a; c)$ and per-context policy $\pi_c(\cdot \mid s) := \pi(\cdot \mid s, c)$.

**Proposition C.2.** *For every contextual policy $\pi(\cdot \mid s, c)$, there is a corresponding per-context policy $\pi_c(\cdot \mid s) = \pi(\cdot \mid s, c)$ such that*

$$
V^\pi(s, c) = \hat{V}^{\pi_c}(s).
$$

*Conversely, for every per-context policy $\pi_c(\cdot \mid s)$ there is a contextual policy $\pi(\cdot \mid s, c) = \pi_c(\cdot \mid s)$ with the same return. Consequently,*

$$
\sup_\pi V^\pi(s, c) = \sup_{\pi_c} \hat{V}^{\pi_c}(s),
$$

*and optimal contextual policies and optimal per-context policies coincide on the reachable set.*

*Proof sketch.* This contextualization with fixed $c$ is an annotated MDP in the sense of (Bacchus et al., 1996, Def. 4.1), with extended states $(s, c)$ and stepwise rewards $R_n^{\text{ctx}}(s, a; c)$. For any $\pi(\cdot \mid s, c)$, the $(s, a)$-trajectory law under $\mathcal{M}_d^{\text{ctx}}$ coincides with that under the per-context policy $\pi_c(\cdot \mid s)$ in $\widehat{\mathcal{M}}_d$; moreover the stepwise rewards agree by construction $R_n^{\text{ctx}}(s, a; c) = \hat{R}_{\pi_c, n}(s, a)$ at the fixed context. Hence $V^\pi(s, c) = \hat{V}^{\pi_c}(s)$ on the reachable set. The projection/lifting correspondence for annotated expansions (cf. (Bacchus et al., 1996, Prop. 4.3 and Cor. 4.4)) then yields equality of suprema and optimal policies on the reachable set. □

This formulation keeps $c$ as an explicit input to the reward while remaining per-episode stationary because $c$ is fixed within an episode. It is thus a notationally different but also equivalent way to present the same optimization problem as in the standard MDP.

# D ALGORITHM AND IMPLEMENTATION

## D.1 SOURCE CODE

We will open-source our code upon acceptance.

## D.2 REWARD FUNCTION DESIGN

For mathematical reasoning tasks, we use binary rewards based on final answer correctness:

$$R(s_T, a_T) = \begin{cases} 1.0 & \text{if final answer is correct} \\ 0.0 & \text{if final answer is incorrect} \end{cases} \quad (6)$$

The reward is only assigned at the final step of each trajectory when the complete solution is generated. This sparse reward structure is typical for mathematical reasoning tasks where intermediate steps cannot be easily evaluated without domain expertise.

## D.3 KL DIVERGENCE COMPUTATION

The KL divergence between student and teacher policies is computed at each time step as:

$$\text{KL}(\pi_\theta(\cdot|s_t)||\mu(\cdot|s_t)) = \sum_{a \in \mathcal{V}} \pi_\theta(a|s_t) \log \frac{\pi_\theta(a|s_t)}{\mu(a|s_t)} \quad (7)$$

where $\mathcal{V}$ is the LLM vocabulary.

## D.4 HYPERPARAMETER SETTINGS

We used the following hyperparameters for all the method:

- Batch size: 64 responses (8 questions $\times$ 8 responses per question)
- Learning rate: $1^{e-5}$
- Optimizer: AdamW
- Discount factor $\gamma = 1$
- Constraint threshold $d = 0.35$. The constraint threshold was selected based on preliminary experiments that seek to minimize only the KL (mini-LLM and GKD).
- Number of training epochs: 20
- Penalty $n$: 20

The training of Llama3.2-3B with GRPO was unstable due to its very poor initial performance; therefore, to bootstrap all methods, we apply KL distillation alone for the first 3 epochs (even with GRPO $\lambda = 0$).

## D.5 TRAINING TIME

The training takes less than 2 days on a single accelerator for each method. Overall, all the methods need the same amount of training time. GRPO is only a bit faster because the teacher is not used, but backward phases and generation time dominate the overall training time.

# E MORE EXPERIMENTS RESULTS

Table 1: Distillation results of Qwen2.5-1B on GSM8K. Higher final answer correctness (FAC), reasoning win rate (RWR) and constraint satisfaction (CS) are better, while lower KL divergence and lower reasoning loose rate (RLR) are better.

| Method | Apple/GSM-Symbolic | | | | | GSM8K | | | | | MATH | | | | |
|---|---|---|---|---|---|---|---|---|---|---|---|---|---|---|---|
| | FAC↑ | RWR↑ | RLR↓ | KL↓ | CS↑ | FAC↑ | RWR↑ | RLR↓ | KL↓ | CS↑ | FAC↑ | RWR↑ | RLR↓ | KL↓ | CS↑ |
| Ours | 70.80 | **60.55** | **10.58** | **0.16** (± **0.17**) | 96.1 | 76.11 | **58.72** | **7.86** | 0.15 (±0.19) | 94.99 | 25.78 | **41.65** | **14.44** | 0.15 (±0.17) | 96.2 |
| GRPO λ = 0.0 | **75.94** | 14.89 | 53.58 | 0.41 (± 0.28) | 41.74 | **80.06** | 12.15 | 54.67 | 0.41 (± 0.29) | 42.83 | **29.04** | 19.49 | 39.62 | 0.27 (±0.19) | 78.68 |
| GKD-GRPO λ = 0.001 | 75.50 | 10.64 | 57.88 | 0.29 (±0.23) | 76.6 | 80.51 | 10.94 | 55.71 | 0.28 (±0.17) | 78.01 | 28.60 | 18.5 | 38.73 | 0.23 (±0.17) | 87.40 |
| GKD-GRPO λ = 0.01 | 72.52 | 34.87 | 25.27 | 0.18 (±0.25) | 94.2 | 75.89 | 34.52 | 23.76 | 0.18 (±0.23) | 92.11 | 29.04 | 26.55 | 24.23 | 0.15 (±0.14) | 94.94 |
| GKD-GRPO λ = 0.1 | 69.88 | 22.34 | 36.04 | 0.16 (±0.23) | 94.92 | 75.13 | 20.82 | 35.36 | **0.14 (± 0.20)** | 94.61 | 27.68 | 20.86 | 30.23 | **0.14** (±0.15) | 95.46 |
| GKD-GRPO λ = 1.0 | 67.47 | 29.12 | 17.74 | 0.17 (±0.29) | 94.34 | 73.69 | 29.01 | 16.37 | 0.16 (± 0.32) | 94.08 | 27.16 | 24.04 | 17.37 | 0.15 (±0.21) | 95.02 |
| GKD-GRPO λ = 10 | 67.8 | 30.01 | 17.59 | 0.16 (±0.25) | 94.66 | 74.07 | 28.94 | 16.73 | 0.15 (±0.23) | 93.1 | 26.6 | 24.01 | 17.82 | 0.15 (±0.18) | 95.12 |
| GKD | 68.34 | 28.1 | 19.24 | 0.16 (±0.25) | 94.88 | 74.37 | 27.03 | 18.18 | 0.15 (±0.23) | 94.08 | 26.42 | 23.12 | 18.07 | 0.15 (±0.17) | 94.98 |
| Mini-LLM | 68.02 | 27.65 | 20.24 | 0.16 (±0.28) | 94.2 | 74.22 | 26.20 | 19.68 | 0.15 (±0.26) | 93.78 | 27.06 | 22.01 | 19.71 | 0.15 (±0.21) | 95.56 |
| Student model | 0 | | | 2.08 (±1.89) | 0.14 | 0.22 | | | 1.96 (±1.82) | 0.45 | 0.54 | | | 2.47 (±2.09) | 3.4 |
| Teacher model | 88.12 | | | | | 92.27 | | | | | 34.46 | | | | |

Table 2: Distillation results of Llama3.2-3B on MATH. Higher success rates (SR) and constraint satisfaction (CS) are better, while lower KL divergence is better.

| Method | Apple/GSM-Symbolic | | | | | GSM8K | | | | | MATH | | | | |
|---|---|---|---|---|---|---|---|---|---|---|---|---|---|---|---|
| | FAC↑ | RWR↑ | RLR↓ | KL↓ | CS↑ | FAC↑ | RWR↑ | RLR↓ | KL↓ | CS↑ | FAC↑ | RWR↑ | RLR↓ | KL↓ | CS↑ |
| Ours | 36.78 | **42.33** | **21.44** | 0.22 (±0.07) | **94.64** | 38.36 | **51.76** | **19.78** | 0.21 (±0.07) | **99.60** | 17.10 | **34.40** | **23.58** | 0.15 (±0.06) | 99.48 |
| GRPO λ = 0.0 | **42.48** | 33.82 | 39.12 | 0.71 (±0.15) | 0.16 | 49.73 | 21.30 | 57.14 | 0.73 (±0.15) | 0.3 | **18.90** | 25.44 | 47.96 | 0.64 (±0.2) | 8.08 |
| GKD-GRPO λ = 0.001 | 40.20 | 38.42 | 32.37 | 0.49 (±0.12) | 14.56 | **53.44** | 37.18 | 34.36 | 0.5 (±0.13) | 12.81 | 18.52 | 33.87 | 34.65 | 0.39 (±0.14) | 38.98 |
| GKD-GRPO λ = 0.01 | 40.22 | 23.81 | 33.77 | 0.29 (±0.09) | 72.86 | 52.53 | 43.60 | 28.22 | 0.28 (±0.09) | 80.89 | 17.62 | 29.21 | 29.32 | 0.21 (±0.08) | 93.52 |
| GKD-GRPO λ = 0.1 | 42.28 | 27.21 | 27.65 | 0.23 (±0.08) | 90.56 | 53.37 | 32.74 | 35.67 | 0.23 (±0.08) | 92.57 | 17.48 | 30.07 | 25.38 | 0.16 (±0.07) | 98.20 |
| GKD-GRPO λ = 1.0 | 38.02 | 24.31 | 30.63 | **0.21 (±0.08)** | 94.18 | 42.45 | 31.99 | 35.24 | 0.21 (±0.07) | 95.98 | 17.80 | 27.38 | 29.90 | **0.14** (±0.06) | 99.22 |
| GKD-GRPO λ = 10 | 37.92 | 26.16 | 28.10 | **0.21 (±0.08)** | 94.5 | 38.66 | 30.08 | 36.90 | **0.20** (±0.07) | 95.60 | 18.42 | 30.17 | 28.18 | **0.14** (±0.06) | 99.46 |
| GKD | 36.88 | 26.87 | 27.71 | **0.21 (±0.08)** | 94.7 | 38.36 | 41.74 | 28.42 | **0.20** (±0.07) | 95.98 | 17.80 | 29.80 | 25.66 | **0.14** (±0.06) | 99.24 |
| Mini-LLM | 37.34 | 26.46 | 28.66 | **0.21 (±0.08)** | 94.38 | 39.25 | 26.12 | 40.73 | 0.21 (±0.07) | 95.52 | 15.88 | 30.27 | 26.22 | **0.14** (±0.06) | 99.20 |
| Student model | 0 | | | 0.73 (±0.88) | | 2.75 | | | 1.45 (±1.57) | | 0.08 | | | 1.02 (±0.9) | |
| Teacher model | 51.86 | | | | | 54.58 | | | | | 32.08 | | | | |

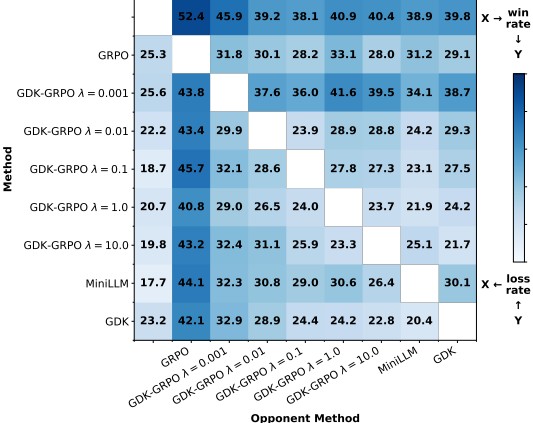

Figure 5: Pairwise comparison heatmap showing the relative performance of our method against baselines, averaged across all three evaluation datasets (Apple/GSM-Symbolic, GSM8K, and MATH) on Llama3.2-3B. Averaging over columns gives the reasoning win rate (RWR) and over rows the reasoning loss rate (RLR).

# F    GENERATED ANSWERS

---

**Question**:
Leilani saw a 300-foot whale with 4 45-inch remoras attached to it. What percentage of the whale's body length is the combined length of the remoras?

---

**Ours**:

To determine what percentage of the whale's body length the combined length of the remoras is, we need to follow these steps:
**1. Convert the length of the remoras from inches to feet:**

$$45 \text{ inches} \times \frac{1 \text{ foot}}{12 \text{ inches}} = 3.75 \text{ feet}$$

So, each remora is 3.75 feet long.
**2. Calculate the total length of the remoras:**

$$4 \text{ remoras} \times 3.75 \text{ feet per remora} = 15 \text{ feet}$$

**3. Determine the total length of the whale:**

$$300 \text{ feet}$$

**4. Calculate the percentage of the whale's body length that the combined length of the remoras represents:**

$$\left(\frac{15 \text{ feet}}{300 \text{ feet}}\right) \times 100\%$$

**5. Simplify the fraction:**

$$\frac{15}{300} = 0.05$$

**6. Convert the decimal to a percentage:**

$$0.05 \times 100\% = 5\%$$

Therefore, the combined length of the remoras is $\boxed{5}$ percent of the whale's body length.

**GRPO $\lambda = 0$:**

Let's break down the problem step by step to find the percentage of the whale's body length that the combined length of the remoras represents.

First, we need to convert all units to the same unit. Let's use feet for this calculation.

The length of each remora is given as 45 inches. Since there are 12 inches in a foot, the length of each remora in feet is:

$$\frac{45}{12} = 3.75 \text{ feet}$$

Since there are 4 remoras, the total length of the remoras is:

$$4 \times 3.75 = 15 \text{ feet}$$

Next, we add the length of the whale to the combined length of the remoras to get the total length of the whale:

$$300 + 15 = 315 \text{ feet}$$

Now, we need to find what percentage the combined length of the remoras is of the whale's body length. This can be calculated using the formula for percentage:

$$\left(\frac{15}{315}\right) \times 100$$

Simplifying the fraction inside the percentage formula:

$$\frac{15}{315} = \frac{1}{21}$$

$$\left(\frac{1}{21}\right) \times 100 = \frac{100}{21} \approx 4.7619$$

Rounding to the nearest whole number, we get:

$$\boxed{5}$$

---

**Judge**:
Response A correctly calculates the percentage by comparing the remoras' total length to the whale's length without adding them, ensuring accuracy. Response B incorrectly adds the remoras' length to the whale's, leading to a flawed calculation.
Verdict: A wins

---

Figure 6: Example of generated answer with Qwen2.5-1.5B-Math after distillation.

**Question**:
Wei saw a 200-foot whale with 2 12-inch remoras attached to it. What percentage of the whale's body length is the combined length of the remoras?

**Ours**:

To determine what percentage of the whale's body length the combined length of the remoras is, we need to follow these steps:
1. **Convert the length of the remoras from inches to feet:**
- Each remora is 12 inches long.
- There are 12 inches in a foot.
- Therefore, the length of each remora in feet is:

$$\frac{12 \text{ inches}}{12 \text{ inches per foot}} = 1 \text{ foot}$$

- Since there are 2 remoras, the total length of the remoras in feet is:

$$2 \text{ remoras} \times 1 \text{ foot per remora} = 2 \text{ feet}$$

2. **Calculate the percentage of the whale's body length:**
- The whale's body length is 200 feet.
- The combined length of the remoras is 2 feet.
- To find the percentage, we use the formula:

$$\text{Percentage} = \left( \frac{\text{Length of remoras}}{\text{Whale's body length}} \right) \times 100$$

- Substituting the values, we get:

$$\text{Percentage} = \left( \frac{2 \text{ feet}}{200 \text{ feet}} \right) \times 100$$

- Simplify the fraction:

$$\frac{2}{200} = 0.01$$

- Multiply by 100 to convert to a percentage:

$$0.01 \times 100 = 1\%$$

Therefore, the combined length of the remoras is $\boxed{1}$ % of the whale's body length.

**GRPO** $\lambda = 1$:

To determine what percentage of the whale's body length the combined length of the remoras is, we need to follow these steps:
1. Convert the length of the remoras from inches to feet.
2. Add the length of the remoras to the length of the whale.
3. Calculate the percentage of the whale's body length that the combined length of the remoras represents.
First, we convert the length of the remoras from inches to feet. Since there are 12 inches in a foot, we divide 12 by 12 to get 1 foot per inch. Therefore, the length of each remora is:

$$12 \text{ inches} \div 12 = 1 \text{ foot}$$

Since there are 2 remoras, the total length of the remoras is:

$$2 \text{ remoras} \times 1 \text{ foot/remora} = 2 \text{ feet}$$

Next, we add the length of the remoras to the length of the whale:

$$200 \text{ feet} + 2 \text{ feet} = 202 \text{ feet}$$

Now, we calculate the percentage of the whale's body length that the combined length of the remoras represents. We do this by dividing the total length of the remoras by the length of the whale and then multiplying by 100:

$$\left( \frac{2 \text{ feet}}{200 \text{ feet}} \right) \times 100 = 1\%$$

Therefore, the combined length of the remoras is $\boxed{1}$ % of the whale's body length.

**Judge**:
Response A is more clear and logically structured, breaking down each step with proper mathematical notation and avoiding unnecessary complexity. Response B incorrectly adds the remoras' length to the whale's length, which is irrelevant to the problem and introduces confusion.
Verdict: A wins

Figure 7: Example of generated answer with Qwen2.5-1.5B-Math after distillation.

**Question**:
There are 154 fourth-graders at Maple Grove School. 80 of them are girls. On Thursday, 3 fourth-grade girls and 6 fourth-grade boys were absent. How many fourth grade boys were at Maple Grove School on Thursday?

**Ours**:

To determine how many fourth-grade boys were at Maple Grove School on Thursday, we need to follow these steps:
1. **Calculate the number of fourth-grade boys:**
- The total number of fourth-graders is 154.
- The number of fourth-grade girls is 80.
- Therefore, the number of fourth-grade boys is:

$$154 - 80 = 74$$

2. **Determine the number of fourth-grade boys absent:**
- On Thursday, 6 fourth-grade boys were absent.
3. **Calculate the number of fourth-grade boys present:**
- The total number of fourth-grade boys is 74.
- The number of fourth-grade boys absent is 6.
- Therefore, the number of fourth-grade boys present is:

$$74 - 6 = 68$$

Thus, the number of fourth-grade boys at Maple Grove School on Thursday is $\boxed{68}$.

**GDK**:

To determine how many fourth-grade boys were at Maple Grove School on Thursday, we need to follow these steps:
1. **Calculate the number of fourth-grade girls present:**
- Total number of fourth-graders: 154
- Number of fourth-grade girls: 80
- Number of fourth-grade girls present: $154 - 3 = 151$
2. **Calculate the number of fourth-grade boys present:**
- Total number of fourth-graders: 154
- Number of fourth-grade boys: $154 - 80 = 74$
- Number of fourth-grade boys present: $74 - 6 = 68$
Thus, the number of fourth-grade boys present on Thursday is $\boxed{68}$.

**Judge**:
Response A provides a clearer and more accurate approach by first determining the total number of boys and then subtracting the absent ones. Response B incorrectly calculates the number of girls present, which could lead to confusion.
Verdict: A wins

Figure 8: Example of generated answer with Qwen2.5-1.5B-Math after distillation.

