# OpenReview forum: "Rethinking Large Language Model Distillation: A Constrained Markov Decision Process Perspective"
_ICLR.cc/2026/Conference — ICLR 2026 Conference Withdrawn Submission_

### Official Review · Reviewer_pe48 · 2025-10-30

**Soundness:** 4
**Presentation:** 4
**Contribution:** 2
**Rating:** 4
**Confidence:** 4

**Summary:**

This paper introduces a reward signal during the distillation phase, formulating the process as a Constrained MDP problem. By eliminating the augmented state, it derives the reward under a KL-threshold constraint and presents the corresponding Policy Gradient optimization objective. The paper provides a proof of the method’s optimality and demonstrates its effectiveness through experiments: it not only preserves the KL constraint and answer accuracy but also improves the quality of reasoning, achieving a good trade-off between correctness and constraint satisfaction.

**Strengths:**

1. This paper presents a new constrained MDP perspective on distillation and simplifies implementation by decomposing the gradient and threshold mechanisms.
2. This paper adapts the Saute method by removing the state augmentation step, ensuring the student model
operates independently of the teacher at test time while maintaining the theoretical guarantees
and enhancing exploration on constraint-violating trajectories.

3. This paper conducts extensive experiments on mathematical reasoning tasks

**Weaknesses:**

1. The conceptual insight seems questionable. Constrained RL is primarily applied in domains such as safety, where violating constraints could lead to severe consequences, for example, a car exceeding the speed limit. In such cases, constraints are necessary because they ensure safe and reliable operation. However, in this paper, it is unclear why the knowledge distillation
 still requires constraints. Slightly exceeding the constraints would not lead to serious consequences, so the necessity of introducing constraints here is not well justified.


2. The chosen metrics seem inappropriate:
In light of the first point, the role of the constraint satisfaction metric is unclear. Moreover, in reasoning tasks, the primary metric should be Final Answer Correctness. However, the improvement in Final Answer Correctness appears marginal. Metrics such as Reasoning Win Rate and Reasoning Loss Rate appear less meaningful in reasoning tasks. Could the authors clarify the motivation and interpretation of these three metrics?

3. This work can be viewed as an adapter-based variant of GKD. I would suggest exploring a soft constraint formulation instead, which might provide better flexibility and alleviate the overly rigid effects introduced by the current constrained setup.

4. Could you provide results using larger models, such as Qwen2.5-14B as the teacher model and Qwen2.5-7B as the smaller model?

**Questions:**

Please see questions in weakness.

---

### Official Review · Reviewer_ezR9 · 2025-10-31

**Soundness:** 1
**Presentation:** 3
**Contribution:** 1
**Rating:** 2
**Confidence:** 4

**Summary:**

This paper considers LLM distillation. Specifically, authors formalize this problem as maximizing task-specific rewards while constraining the divergence from the teacher model to remain below a specified threshold. Extensive experiments demonstrate the effectiveness of the proposed method.

**Strengths:**

1. This paper is clearly written and easy to follow.

**Weaknesses:**

1. Limited technical novelty. Authors claim that their approach "introducing a modified reward function that
maintains theoretical guarantees of constraint satisfaction without requiring state augmentation or teacher model access during deployment and without the computational overhead of the dual Lagrangian methods". However, from my perspective, the proposed framework is nothing but dual Lagrangian method. Specifically, the proposed objective shown in Equation (5) is indeed the deriviative of the objective as shown in line 174, with $\lambda$ being a fixed hyper-parameter.
2. GSM8K is easy. Maybe authors can consider harder benchmarks.

**Questions:**

Please as Weaknesses above.

---

### Official Review · Reviewer_tDpb · 2025-11-06

**Soundness:** 2
**Presentation:** 3
**Contribution:** 2
**Rating:** 4
**Confidence:** 3

**Summary:**

This paper introduces a novel approach to LLM distillation by formulating it as a constrained reinforcement learning (CRL) problem, aiming to maximize task-specific rewards (like final answer correctness) while strictly constraining the student model's divergence from the teacher model. The authors adapt a CRL method Saute but remove its state-augmentation step, which normally tracks the remaining budget. This modification creates a new reward function that penalizes constraint violations while still providing informative feedback, crucially allowing the student model to operate without needing the teacher at test time. Experiments on mathematical reasoning tasks demonstrate that this method achieves a superior balance, maintaining high constraint satisfaction and reasoning quality while preserving competitive task performance compared to baselines that optimize for rewards or divergence alone.

**Strengths:**

- By adopting CRL, the method mitigates the need to tune the hyperparameter $\lambda$, while still integrating the external reward.

- The approach leverages deterministic state transitions in LLMs to remove the need for state augmentation when solving the CRL optimization problem.

- Theoretical analysis demonstrates that the reformulated objective preserves the constraint satisfaction guarantees.

- Experimental results show that the method effectively balances the final score and divergence from the teacher model, producing high-quality reasoning in model responses.

- Experiments on models from different families further demonstrate the algorithm’s generalizability across architectures.

**Weaknesses:**

- The main motivation for CRL compared with GKD-GRPO is that *tuning the hyperparameter $\lambda$ is challenging*. This claim would be stronger if supported by references, motivating experiments, or further explanation.

- The reward design includes a policy-dependent discrepancy term $\Phi$. While this is intuitively useful, an ablation study would help demonstrate the necessity and contribution of this term.

- In Figure 3, the proposed method performs comparably to GKD with $\lambda=0.01$. Is it fair to conclude that CRL does not yield a noticeable performance gain over GKD with $\lambda=0.01$? If so, what is the main advantage of CRL over GKD?

- The implementation details for evaluating Reasoning Quality are not provided. For instance, it would be helpful to specify the prompt template used for the judge and clarify whether the order of model responses in the prompt affects the judgment.

- Typos and minor corrections:

  Figure 4 caption: “3B” → “1.5B”

  Table 1 caption: “1B” → “1.5B”; “GSM8K” → “all benchmarks”

  Table 2 caption: “MATH” → “all benchmarks”

**Questions:**

In Figure 4, CRL outperforms other approaches in reasoning quality. However, Figure 2 and 3 show that GKD and mini-LLM exhibit similar KL divergence from the teacher model. Could the authors elaborate on why CRL achieves superior reasoning quality while GKD and mini-LLM, despite having comparable KL distances, do not?

---

### Note · Authors · 2025-11-12

I have read and agree with the venue's withdrawal policy on behalf of myself and my co-authors.